Mining of co-expression genes in response to cold stress at maize (Zea mays L.) germination and sprouting stages by weighted gene co-expression networks analysis

Shi Chuangye 1 2
Dong Jing 2
Zhang Chunxiao 2
Sun Liquan 1
Jin Fengxue 2
Zhou Xiaohui 2
Liu Xueyan 2
Wu Weilin 1 wlwu@ybu.edu.cn
Li Xiaohui 2 lixiaohui2002lix@163.com
1 Agricultural College, Yanbian University , Yanji, Jilin Province , China
2 Maize Research Institute, Jilin Academy of Agricultural Sciences , Gongzhuling, Jilin Province , China
Kutlu Imren
Electronic publication date: 2025 Mar 11
Publication date: 2025
Volume: 13
Electronic Location ID: e19124
Received 2024 Dec 9; Accepted 2025 Feb 17
Copyright: © 2025 Shi et al.
Copyright year: 2025
Copyright holder: Shi et al.
License: This is an open access article distributed under the terms of the Creative Commons Attribution License, which permits unrestricted use, distribution, reproduction and adaptation in any medium and for any purpose provided that it is properly attributed. For attribution, the original author(s), title, publication source (PeerJ) and either DOI or URL of the article must be cited.
License URL: https://creativecommons.org/licenses/by/4.0/

Keywords: Maize, Weighted gene co-expression network, Transcriptome, Hub genes, Low temperature stress

Funding: The Agricultural Science and Technology Innovation Program of Jilin Province CXGC2023RCY047 Major Science and Technology Projects in Biological Breeding 2023ZD04067 This project was supported by “The Agricultural Science and Technology Innovation Program of Jilin Province” (No. CXGC2023RCY047) and Major Science and Technology Projects in Biological Breeding (No. 2023ZD04067). The funders had no role in study design, data collection and analysis, decision to publish, or preparation of the manuscript.

==============================
Background

Maize (Zea mays L.) is one of the main agricultural crops with the largest yield and acreage worldwide. Maize at the germination and sprouting stages are highly sensitive to low-temperatures, especially in high-latitude and high-altitude regions. Low-temperature damage in early spring presents a major meteorological disaster in maize, severely affecting plant growth and maize yield. Therefore, mining genes tolerant to low temperatures is crucial. We aimed to analyze differential gene expression and construct co-expression networks in maize under low temperatures.

Methods

Inbred lines, Zhongxi 091/O2 and Chang 7-2, are tolerant and sensitive to low temperatures at the germination and sprouting stages, respectively. We grew these lines at 10 °C and 2 °C at the germination and sprouting stages, respectively. Samples were taken at five time points (0, 6, 12, 24, and 36 h) during the two stages, and transcriptome sequencing was performed. The analyses were conducted using weighted gene co-expression networks analysis (WGCNA), Gene Ontology (GO), the Kyoto Encyclopedia of Genes and Genomes (KEGG), and gene co-expression networks.

Results

WGCNA was used to construct co-expression networks at two stages, resulting in six and nine co-expression modules, respectively. Two modules at the germination stage (blue and yellow) and two modules at the sprouting stage (turquoise and magenta) were identified. These were significantly associated (p < 0.01) with tolerance at low temperature. The differentially expressed genes (DEGs) in the four modules revealed entries related to hormone and oxygen-containing compound responses by GO functional enrichment. Among the four modules, DEGs from three modules were all significantly enriched in the MAPK signaling pathway. Based on the connectivity, the top 50 genes for each module were selected to construct a protein interaction network. Seven genes have been proven to be involved in the response to low-temperature stress.

Conclusion

WGCNA revealed the differences in the response patterns of genes to low-temperature stress between tolerant and sensitive lines at different time points. Seven genes involved in low-temperature stress were functionally annotated. This finding suggests that WGCNA is a viable approach for gene mining. The current findings provide experimental support for further investigation of the molecular mechanisms underlying tolerance to low temperatures in maize.

Introduction

Maize (Zea mays L.) is a major food, economic and feed crop worldwide (Wang et al., 2020). Due to its origin in tropical and subtropical regions, the growth and development of maize are highly sensitive to low-temperatures (Greaves, 1996). When planting in high latitude or high-altitude areas, the sudden low-temperature damage at the germination and sprouting stages can lead to cell membrane disorders, protein denaturation, metabolic dysfunction, and changes in oxidative defense and osmotic pressure, further resulting in irreversible damage to cells and tissues (Allen & Ort, 2001; Hussain et al., 2018; Nguyen et al., 2009). This not only reduces seed germination rates and seedling vigor, but also increases the chances of soil bacteria infecting maize (Juurakko, Cenzo & Walker, 2021). Hence, identifying genes capable of enhancing cold tolerance is crucial for low-temperature adaptation, ultimately ensuring the yield stability.

There have been some reports on gene mining tolerant to low-temperature stress in maize. Low-temperature stress induces the over-expression of four genes (ZmDREB1A, ZmDREB2A, ZmDBP3, and ZmDBP4), which enhances cold tolerance in maize (Qin et al., 2004). ZmICE1 directly regulates the expression of DREB1. Natural variations in the ZmICE1 promoter affect the binding affinity with the transcriptional activator ZmMYB39, which is a positive regulator of cold tolerance in maize. ZmMYB39 results in varying levels of ZmICE1 transcription and cold tolerance among inbred lines (Jiang et al., 2022). ZmMYB31 reduces ion leakage, ROS levels, and low-temperature photo inhibition by positively regulating the expression of genes related to low-temperature stress, thereby enhancing cold tolerance (Li et al., 2019). In maize leaves, ZmMPK5 was found to be involved in plant recovery from low temperature stress (Berberich, Sano & Kusano, 1999). Kong et al. (2011) isolated the group C MAPKK gene ZmMKK4 from the root system of the cultivar Zhengdan958 and found that the expression of the ZmMKK4 transcript was up-regulated due to low-temperature exposure. A recent study identified a novel cold regulation pathway in maize, ZmMPK8-ZmRR1-ZmDREB1.10/ZmCesA2. ZmRR1 accumulates and induces the expression of ZmDREB1s and ZmCesAs, thereby enhancing cold tolerance. ZmMPK8 is a negative regulator of cold tolerance that interacts with ZmRR1 and phosphorylates ZmRR1 at Ser15. A natural variation of ZmRR1 with a 45-bp deletion that encompasses Ser15 prevents its phosphorylation by ZmMPK8 (Zeng et al., 2021).

With the rapid development of high-throughput sequencing technologies, large amounts of sequencing data have been generated. WGCNA is based on transcriptome sequencing data. According to the difference in gene expression level, the genes are divided into different categories. If the gene expression patterns are similar, they are divided into the same module, the saliency module related to the research trait is selected, and the key genes in the module are screened. Since Langfelder & Horvath (2008) proposed WGCNA, several key genes have been identified by this method, which are relevant to plant phenotypic traits, responses to biotic or abiotic stresses, and other aspects. Zhang & Horvath (2005) found 13 modules significantly correlated with corn kernel weight, with a correlation range of −0.87 to 0.89. Yu et al. (2023) used B144 (cold stress tolerant) and Q319 (cold stress sensitive) as experimental lines and identified four specific modules (deep orange, green-yellow, light yellow and purple) related to low-temperature tolerance at the maize seedling stage by WGCNA. Wang et al. (2024) identified five hub genes directly related to salt response from two alfalfa (Medicago sativa L) varieties with different resistances by WGCNA. Zhao et al. (2024) screened 11 co-expression modules closely related to cold stress resistance in quinoa (Chenopodium quinoa Wild.) seedlings and identified four important transcription factors involved in resisting cold stress from two of the modules. Liang et al. (2022) found four hub genes involved in Ca2+ transport in maize in response to salt stress. Zhao et al. (2023) determined 19 and 49 hub genes that regulated maize mesocotyl and coleoptile elongation plasticity under different light spectral quality stimulation.

We aimed to conduct differential expression analysis of transcriptome data of maize subjected to the 10 °C-germination stage and 2 °C-sprouting stage using antagonistic and sensory lines at different times. WGCNA was used to construct gene co-expression networks, linking gene expression modules with cryogenic treatments. Enrichment analysis was conducted to explore the function of differentially expressed core gene co-expression networks at the germination and sprouting stage of maize under low-temperature stress, which provided new clues and insights for further research on the molecular mechanism of maize low temperature stress.

Materials and Methods

Experimental lines and treatments

In this experiment, different degrees of cold treatment were applied at two maize germination stages. The maize inbred lines Zhongxi 091/O2 and Chang 7-2 were subjected to RNA sequencing, which carried out by the Maize Research Institute of the Jilin Academy of Agricultural Sciences in Jilin, China. The maize inbred lines Zhongxi 091/O2 and Chang 7-2 were found to be resistant (R) and susceptible (S) (Liu et al., 2021) to cold stress at the germination and sprouting stages in previous studies, respectively. The identification data of the two lines are detailed in Fig. S1.

Germination stage: 200 plump seeds were selected and disinfected with 0.1% HgCl2 for 10 min. The seeds were soaked at 25 °C for 12 h and then subjected to low-temperature treatment at 10 °C. The seeds were placed on a disinfected culture dish covered with a moist sponge and filter paper at a density of 20 seeds per dish. The dishes were placed separately in a smart light incubator (10 °C, 24 h dark culture, and 60% humidity). Samples were collected at five time intervals (0, 6, 12, 24, and 36 h), and several neatly germinating seeds were randomly selected. The embryos were dissected using a dissecting needle and weighed to a total weight of 0.6 g. The experiment was repeated three times to ensure accuracy.

Sprouting stage: 500 plump seeds were selected and disinfected with 0.1% HgCl2 for 10 min. The seeds were placed on a sterilized culture dish covered with a moist sponge and filter paper at a density of 20 seeds per dish. The dishes were placed in a smart light incubator for germination (25 °C, 24 h dark culture, and 60% humidity). When the sprouts were approximately 4 cm long, 100 seedlings of the same growth size were selected and subjected to low-temperature treatment at 2 °C. The samples were collected at five intervals (0, 6, 12, 24, and 36 h) and several seeds with the same sprout lengths were randomly selected. The seedlings were cut into cubes and weighed for a total weight of 0.6 g. The experiment was repeated three times to ensure accuracy.

The total RNA of maize at the germination and sprouting stages was extracted using the polysaccharide polyphenol plant total RNA extraction kit (Exp.01032020, stored at RT) of the Shanghai Pudi Biotechnology Co., Ltd. After purification, the ends were repaired, base A was added to the 3′ end, and then the sequencing adapter sequence was added. The target size fragment after gel electrophoresis was recovered, the library required for sequencing was amplified by PCR, and the sequencing was completed using an Illumina HiSeq™ 2000 sequencer.

Clustering analysis and functional enrichment of module genes

Gene co-expression network analysis was performed using the WGCNA package in R software (https://cran.r-project.org/web/packages/WGCNA/index.html). To ensure compliance with scale-free network distributions, for WGCNA, the appropriate weighting coefficient β must be selected. The value of coefficient β was estimated using the pick Soft Threshold function in the WGCNA package. First, the β = 1–30 was set to calculate the corresponding correlation coefficient and mean gene connectivity. The selection criterion of β is to ensure that the square of the correlation coefficient is as close as possible to 0.8, while also preserving a sufficient level of gene connectivity. The dynamic tree cut method was used to identify the co-expression module. The Automatic Network Builder Block-wise Module was used to build the network. Here, the minimum module size was 30, merge Cut Height = 0.25 merged the modules with a similarity of 0.75, and other parameters were set by default. To obtain the biological functions and signaling pathways involved in each module, genes in each module were subjected to GO analysis (http://systemsbiology.cau.edu.cn/agriGOv2/index.php) and KEGG analysis. The R package was then used to plot and display the results. The threshold for the GO term and the KEGG pathway was set as p < 0.05.

Weighted gene co-expression network analysis and gene network visualization

The gene co-expression network obtained from WGCNA was processed, and Cytoscape V3.6.1 was used to screen hub genes and visualize their network. Each node in the network represents a gene, and the edges represent the relationships between genes.

Real-time quantitative reverse transcription PCR (qRT-PCR) verification

Ten cold tolerance-related genes were randomly selected for qRT-PCR analysis. Total RNA was extracted from maize during the germination and sprouting stages at low temperatures using a Polysaccharide Polyphenol Plant Total RNA Extraction Kit (Shanghai Pudi Biotechnology Company, Shanghai, China). Reverse transcription was performed using the TaKaRa Reverse Transcriptase M-MLV (RNase H) reverse transcription kit (TaKaRa Bio Inc., Shiga, Japan). qRT-PCR was performed on a Quant Studio six and seven real-time fluorescence quantitative PCR instrument (Applied Biosystems, Waltham, MA, USA) using SYBR® Green Real-time PCR Master Mix (Toyobo, Osaka, Japan), following the manufacturer’s instructions. The reaction conditions were as follows: 95 °C for 5 min, 95 °C for 15 s, 58 °C for 30 s, and 70 °C for 30 s, for a total of 35 cycles. Specific primers were designed using Primer 5.0 software and are listed in Table 1 with corn UBI as the internal reference gene. Gene expression levels were calculated using the 2−ΔΔCT method, where ΔCT refers to the difference of the CT value between the target and the internal reference gene (Livak & Schmittgen, 2001).

Table 1 Genes and primer pairs for fluorescence quantitative PCR.

Gene ID	Primer sequence	Description	
Zm00001d034099	ATGTTGATGCCCAAAGACC	Ferredoxin--NADP reductase leaf isozyme 1 chloroplastic	
AGGAAGAGCCAGCCCAGA	
Zm00001d022125	AAGAGGAGATTCGGAGTGG	Exportin-T	
TTAAAGCCCTTGCTGACC	
Zm00001d045557	TGCCTGACTCGATACGCC	Putative cellulose synthase-like family protein	
GCTCCAGCCTTCTTGTTGTG	
Zm00001d039718	ACCTCTTCCGCTCCTTCCTC	Mini-chromosome maintenance complex-binding protein	
TCACCTTCCTTCGGCTTCTT	
Zm00001d003601	ATCATTGGCATCGGGAGT	Auxin response factor 6	
GCTTTGGGATTCAAGAGGT	
Zm00001d028401	TGGACGGCGTACCTCA	Auxin transporter-like protein 3	
GTGCCCTCCGAAAGTG	
Zm00001d030310	CAGCACGCACAAGCAGG	auxin import carrier1	
CCGAACGCCCAGTAGGA	
Zm00001d010863	TTCATTGGCTCAGTCTATTTC	Protein transport inhibitor response 1	
CTTCCTGTTCCGAGTTCTAC	
Zm00001d002533	AGCCAGGCGAGCCAACA	Abscisic acid receptor PYL9	
CCGACGAGACCCACCATCT	
Zm00001d047705	TGACCACCGTCCACCCGTCC	Cyclase/dehydrase family protein	
GGCTGCGAGTGCGAGCTTCT	

Results

RNA sequencing

A total of 230.4 and 228.3 Gb raw readings were obtained from 60 samples (2 varieties × 5 time intervals × 3 replicates) at the germination and sprouting stages, respectively. After filtering, 1,544,300,100 and 1,531,483,946 clean reads were obtained, with a Q30 percentage above 95.00%. The percentage of GC ranged from 54.04% to 57.35%, indicating that all experimental samples were highly reliable in terms of collection and sequencing results. The statistical power of this experimental design, calculated using RNA-Seq Power, was 0.84. To verify the effectiveness of cold stress treatment, the reported marker genes involved in cold stress were evaluated for their expression patterns before and after cold stress treatment, and the results showed that with increasing cold stress duration, the expression of ZmG6PDH5 (LOC100383421) increased and that of ZmDREB1A (LOC103647602) decreased, reaching the lowest value at 24 h. The results of the RNA-seq data were similar to previously reported results, indicating that the cold stress treatment carried out in this study was effective and further verified the reliability of the RNA-seq data (Fig. 1).

Figure 1 Expression patterns of cold stress response marker genes.

(A) Expression of ZmG6PDH5 gene in GR, GS, SR and SS. (B) Expression of ZmDREB1A gene in GR, GS, SR and SS. Abbreviations: Resistant inbred lines at the Germination stage (GR); Susceptible inbred lines at the Germination stage (GS); Resistant inbred lines at the Sprouting stage (SR); Susceptible inbred lines at the Sprouting stage (SS).

Principal coordinates analysis of cold stress

Principal coordinates analysis (PCA) was performed to explore the correlation between sample processing methods. In the PCA of cold stress at the two stages, the first principal coordinate primarily explained the difference in stage expression and had a large contribution, accounting for 62.96% of the total variance. This indicates that the germination and sprouting stage of stress should be analyzed separately when using WGCNA to construct a gene expression network to avoid stage differences in the stress treatment (Fig. 2A).

Figure 2 Principle component analysis of cold stress.

(A) Clustering of cold-resistant inbred line Zhongxi 091/O2 (R) and cold-sensitive inbred line Chang 7-2 (S). (B, C) Clustering of cold-resistant inbred line Zhongxi 091/O2 (R) and cold-sensitive inbred line Chang 7-2 (S) during the germination (G) and sprouting (S) stages. Abbreviations: Resistant inbred lines at the Germination stage (GR); Susceptible inbred lines at the Germination stage (GS); Resistant inbred lines at the Sprouting stage (SR); Susceptible inbred lines at the Sprouting stage (SS).

In the PCA of cold stress of the lines at the germination stage, the first principal coordinate primarily explained the differences in line expression, accounting for 59.96% of the total variance. In the PCA of cold stress of the lines at the sprouting stage, the first principal coordinate primarily explained the difference in line expression and had a large contribution, accounting for 58.67% of the total variance. This indicates that when using WGCNA to construct a gene expression network, the R and S lines under stress at the germination and sprouting stage should be analyzed separately to avoid line differences in the stress treatment (Figs. 2B and 2C).

Differentially expressed genes analysis

There were 546 genes that responded at the germination and sprouting stages of both lines (Fig. 3A). A total of 14,384, 7,301, 10,295 and 10,799 DEGs were observed in Resistant inbred lines at the Germination stage (GR), Susceptible inbred lines at the Germination stage (GS), Resistant inbred lines at the Sprouting stage (SR), and Susceptible inbred lines at the Sprouting stage (SS) within 36 h of low temperature stress, respectively. The number of DEGs for the R and S lines both exhibited an increasing trend with the extension of stress duration, except for SS. SS decreased after reaching its maximum value at 24 h, while the other three treatments reached their maximum at 36 h (Fig. 3B). Compared to the sensitive line Chang 7-2 (18,100), a greater number of genes were involved in Zhongxi 091/O2 (24,679). Notably, the number of up-regulated genes (14,182) was significantly higher than that of down-regulated genes (10,497) in Zhongxi 091/O2. Conversely, there was no significant difference in the number of up-regulated (9,449) and down-regulated genes (8,651) in Chang 7-2 (Figs. 3C and 3D).

Figure 3 Differentially expressed genes (DEGs) analysis.

(A) Venn diagram of DEGs during the germination (G) and sprouting (S) stages of Zhongxi 091/O2 (R) and Chang 7-2 (S). (B) Trend of the number of DEGs changing with treatment time. (C, D) The number of up-regulated and down-regulated DEGs at each time point. Abbreviations: Resistant inbred lines at the Germination stage (GR); Susceptible inbred lines at the Germination stage (GS); Resistant inbred lines at the Sprouting stage (SR); Susceptible inbred lines at the Sprouting stage (SS).

WGCNA analysis

The weight values were calculated, and the appropriate soft threshold was selected to ensure that the co-expression network followed a scale-free network distribution. The results showed that the optimal soft threshold values were β = 20 at the germination stage and β = 15 at the sprouting stage. This was further used to construct a co-expression network (Fig. S2).

First, cluster analysis was carried out according to the expression of genes, and genes with a high degree of clustering were divided into a module. Then, the dynamic cleavage method was used to identify the co-expression modules. Different modules are indicated by different colors, and grey represents the genes not assigned to any module (Fig. 4). Six modules were obtained at the germination stage, among which the turquoise module contained the greatest number of genes (3,635) and the red contained the least number (43). At the sprouting stage, nine modules were formed. The turquoise module had 2,668 genes, whereas the magenta and pink had only 202 genes each (Figs. 4A and 4B).

Figure 4 Gene clustering tree and module cutting, module-trait-relationship heatmap.

(A, B) Gene clustering trees and module partitioning at germination (G) and sprouting (S) stages. (C, D) Module-trait-relationship heatmap at germination (G) and sprouting (S) stages. Abbreviations: Resistant inbred lines at the Germination stage (GR); Susceptible inbred lines at the Germination stage (GS); Resistant inbred lines at the Sprouting stage (SR); Susceptible inbred lines at the Sprouting stage (SS).

At the germination stage, the correlation of the blue module with stress duration changed from significantly negative to significantly positive and was the most significant in Zhongxi 091/O2 at 36 h (r = 0.45, p = 0.01). A significantly positive correlation was observed between the yellow module and stress duration in Zhongxi 091/O2, which reached a maximum at 36 h (r = 0.55, p = 0.002), whereas Chang 7-2 showed no significant correlation with stress duration (Fig. 4C).

At the sprouting stage, significantly positive correlations between the turquoise and magenta modules and stress duration in Zhongxi 091/O2 were observed, which reached a maximum at 36 h (r = 0.66, p < 0.001; r = 0.68, p < 0.001). Finally, we determined that the blue and yellow modules at the germination stage and the turquoise and magenta modules at the sprouting stage were maize low-temperature-specific modules (Fig. 4D).

Enrichment analysis of specific modules under low-temperature stress

To investigate gene function within specific modules under low-temperature stress, the blue and yellow modules at the germination stage and the turquoise and magenta modules at the sprouting stage were annotated. All four modules were enriched in the response to hormone (GO:0009725) and oxygen-containing compounds (GO:1901700).

In the blue module, the genes were mainly enriched in the following: biological processes such as plant organ development (GO:0099402), developmental processes involved in reproduction (GO:0003006), and post-embryonic development (GO:0009791); molecular functions such as transcription factor activity, sequence-specific DNA binding (GO:0003700), nucleic acid binding transcription factor activity (GO:0001071), DNA binding (GO:0003677), and 2-alkenal reductase (NAD(P)+) activity (GO:0032440); and cellular components such as DNA packaging complex (GO:0044815), organelle membrane (GO:0031090), and plasma membrane (GO:0005886) (Fig. 5A). In the yellow module, the genes were mainly enriched in biological processes such as response to abscisic acid (GO:0009737), response to cold (GO:0009409), and hormone-mediated signaling pathways (GO:0009755) (Fig. 5B).

Figure 5 Gene ontology (GO) enrichment analysis of DEGs in module.

(A) Blue module. (B) Yellow module. (C) Turquoise module. (D) Magenta module.

In the turquoise module, the genes were mainly enriched in the following: biological processes such as the abscisic acid activated signaling pathway (GO:0009738), response to jasmonic acid (GO:0009753), and response to cold (GO:0009409); molecular functions such as transcription factor activity, sequence-specific DNA binding (GO:0003700), nucleic acid-binding transcription factor activity (GO:0001071), and protein serine/threonine kinase activity (GO:0004674); and cellular components such as plasma membrane (GO:0005886), cell periphery (GO:0071944), and cell part (GO:0044464) (Fig. 5C).

In the magenta module, the genes were mainly enriched in biological processes such as regulation of cellular macromolecule biosynthetic process (GO:2000112), regulation of transcription, DNA-templated (GO:0006355) and regulation of nucleic acid-templated transcription (GO:1903506), and molecular functions such as nucleic acid binding transcription factor activity (GO:0001071), sequence-specific DNA binding (GO:0003700), zinc ion binding (GO:0008270), and transition metal ion binding (GO:0046914) (Fig. 5D).

KEGG enrichment of the four specific modules was carried out. The blue module included plant hormone signal transduction, quorum sensing, and carbon fixation in photosynthetic organisms. The yellow module included plant hormone signal transduction, mitogen-activated protein kinase (MAPK) signaling pathway-plant, and vitamin B6 metabolism. The turquoise module included the MAPK signaling pathway, plant hormone signal transduction, and phenylpropanoid biosynthesis. The magenta module included plant hormone signal transduction, plant MAPK signaling pathway, and galactose metabolism (Fig. 6).

Figure 6 KEGG pathway analysis of DEGs in module.

(A) Blue module. (B) Yellow module. (C) Turquoise module. (D) Magenta module.

Gene network visualization

Hub genes are typically those that have a high connectivity within a module. We selected the top 50 genes with the highest connectivity in the blue, yellow, magenta, and turquoise modules for mapping and selected the top 10 genes in terms of connectivity as hub genes. We visualized the hub genes and their associated genes and constructed a gene interaction network (Figs. S3–S5). Forty key genes including eight transcription factors tolerant to cold were predicted. Seven genes have been proven to be involved in the response to low-temperature stress (Table S1).

RT verification

qRT-PCR validation was performed for 10 genes (Fig. 7). We found that these genes have differences in their expression profiles in different lines and at different stages. The results showed that 10 candidate genes have a trend from low to high expression. At the germination stage, the expression of nine genes (except Zm00001d028401), was higher in the R line than in the S line, Zm00001d010863 and Zm00001d028401 had the highest expression at 24 h of stress, Zm00001d034099 and Zm00001d045557 had the highest expression at 12 h of stress, and the rest of the six genes all had the highest expression at 36 h. At the sprouting stage, the expression levels of Zm00001d002533, Zm00001d010863, Zm00001d022125, and Zm00001d028401 were significantly higher in the R than in the S lines. The expression of Zm00001d034099 was significantly higher in the S than in the R lines, and the remaining genes were not significantly different in terms of expression between the two lines. Compared with the germination stage, the gene expression at the sprouting stage showed no significant change across time. Overall, 10 genes exhibited similar expression patterns in qRT-PCR and RNA-seq experiments.

Figure 7 (A–J) qRT-PCR validation of DEGs from RNA-seq data.

Abbreviations: Resistant inbred lines at the Germination stage (GR); Susceptible inbred lines at the Germination stage (GS); Resistant inbred lines at the Sprouting stage (SR); Susceptible inbred lines at the Sprouting stage (SS).

Discussion

WGCNA provides new ideas and methods for resolving gene regulatory networks for different traits and has been widely used. In this study, we used WGCNA to identify four specific modules (blue, yellow, turquoise and magenta) for low-temperature tolerance in maize at the germination and sprouting stages. Through KEGG-pathway analysis of the modules and by calculating the characterized genes in these modules, we identified pathways and genes in maize that are involved in the cold stress response. The results provide new clues for further research on the molecular mechanism underlying the tolerance of maize to low-temperature stress.

KEGG pathway enrichment analysis revealed that phytohormone signaling pathways with important roles in abiotic stress were enriched in all four modules of both stages. Abscisic acid is involved in the regulation of responses to abiotic stresses including cold (Qin et al., 2021). Gibberellic acid is also known to play a role in plant cold tolerance and increasing endogenous biosynthesis (Eremina, Rozhon & Poppenberger, 2016; Rihan, Al-Issawi & Fuller, 2017). Salicylic acid (SA) was strongly correlated with an increase in antioxidant enzyme activities in maize seeds during seedling growth at low temperatures (Wang et al., 2013). Exogenous application of brassinolide also increases the germination rate, reduces cold damage to maize seedlings, and increases antioxidant enzyme activity (Sun et al., 2020). The yellow, turquoise, and magenta modules were enriched in the MAPK signaling pathway. MAPKs are involved in several important processes, including stress signaling and development (Kong et al., 2013). The MAPK cascade pathway is a ubiquitous signaling module in eukaryotes that is regulated by changes in hormone and calcium levels. When plants experience cold stress, cytoplasmic calcium levels increase and are sensed by calcium-binding proteins, which interact with other proteins and initiate phosphorylation cascades involving major stress-responsive genes and transcription factors, leading to cold-stress tolerance. Thus, both stages may respond to cold stress by transducing cold signals via phytohormones and MAPKs (Pagter et al., 2017; Xing et al., 2024). Each module all had specifical pathway enrichment as follows: the blue module in quorum sensing, the phosphoinositide 3-kinase-protein kinase B signaling pathway, carbon fixation in photosynthetic organisms, ansamycin biosynthesis, and monoterpene biosynthesis; the yellow module in vitamin B6 metabolism, carotenoid biosynthesis, and glutathione metabolic pathways; the turquoise module in phenylacetone biosynthesis, the Hippo signaling pathway, Drosophila melanogaster, and floral turquoise pigment biosynthesis; and the magenta module in galactose metabolism, pyrimidine metabolism, starch and sucrose metabolism, and glycolysis/glycogenesis pathways, which play important roles in cold stress (Pang et al., 2021).

Of the top 10 genes annotated to function in the four modules, seven have been shown to be associated with cold tolerance. Of these, the genes LOC100383301 and LOC100284949 have been cloned and validated in maize. LOC100383301 has been annotated with calcium-dependent protein kinase family proteins, which play important roles in abiotic stress tolerance through hormone and reactive oxygen species (ROS) signaling (Cheng et al., 2002; Ranty et al., 2016). LOC100284949 has been annotated with the bZIP transcription factor superfamily protein, which plays multiple roles in abiotic stress responses (Baillo et al., 2019; He et al., 2024; Li et al., 2020). The functions of LOC100286109, LOC100191562, LOC100273668, LOC100282914, and LOC100276374 in other crops were validated. LOC100286109 has been annotated with dehydration-responsive element-binding protein 2A (DREB2A) (Ohama et al., 2017). DREB2A is a key transcription factor that activates the expression of several stress-inducible genes in response to both heat and drought conditions by specifically binding to the cis-acting dehydration-responsive element/C-repeat, thereby enhancing plant tolerance to these conditions (Liu et al., 1998; Sakuma et al., 2006). LOC100191562 annotated cytochrome P450s (CYPs) not only are involved in the biosynthesis of phytohormones and secondary metabolites but also accumulate in response to cold stress (Chopra et al., 2015; Wang et al., 2009). LOC100273668 annotates acyl-CoA binding domain-containing proteins (ACBPs), which are involved in lipid metabolism, plant development, and abiotic stress responses (Xiao & Chye, 2009; Xiao & Chye, 2011; Yurchenko & Weselake, 2011). LOC100282914 has been annotated with the RING zinc finger structural domain superfamily of proteins, RING-containing proteins that are E3 ubiquitin ligases (Li et al., 2022), with ubiquitinated target proteins that function in the regulation of a wide range of cellular activities, including cofactor signaling, defense signaling, and a inbred line of abiotic stress responses (Li et al., 2011; Ma et al., 2009; Yang et al., 2019). LOC100276374 annotates the WD40 family of proteins, which enable plants to respond to abiotic stress by reducing ROS and MDA levels (Tian et al., 2023; Xu et al., 2019; Zhang et al., 2018; Zhang et al., 2017). Although WGCNA was unable to truly elucidate mechanisms of genes tolerant to low-temperatures in maize at the germination and sprouting stages, it provide important clues for further research in the future.

Conclusions

We constructed a co-expression network of weighted genes related to low-temperature tolerance traits in maize. Abiotic stress regulatory pathways were found to be closely linked to some of the hub genes. These include the bZIP transcription factor superfamily protein (LOC100284949), the dehydration-responsive element-binding protein 2A (LOC100286109), the cytochrome P450 superfamily protein (LOC100191562), and the acyl-CoA binding protein pseudogene (LOC100285298). However, it is currently unclear what specific roles these genes play in the process. These results provide clues for studying the molecular mechanism underlying low-temperature tolerance in maize seedlings and provide theoretical support for the cultivation of new low-temperature-tolerant maize inbred lines.

Supplemental Information

Supplemental Information 1 Comparison of relative germination percentage and germination survival rate of Zhongxi 091/O2 and Chang 7-2 (p < 0.001).

(A) Relative germination percentage. (B) Germination survival rate.

Supplemental Information 2 Soft threshold determination of gene co-expression network.

(A) Germination (G) stage. (B) Sprouting (S) stage.

Supplemental Information 3 Visualization of the interactions of the top 50 genes with the highest connectivity in the blue module.

Note: (Blue Bubble: The top 50 genes in module connectivity. Purple Bubble: The top 10 genes in module connectivity. Green Bubble: The genes proven to be involved in low-temperature stress among the top 10 genes in module connectivity. Pink Bubble: Transcription factors among the top 10 genes in module connectivity.)

Supplemental Information 4 Visualization of the interactions of the top 50 genes with the highest connectivity in the yellow module.

Note: (Blue Bubble: The top 50 genes in module connectivity. Purple Bubble: The top 10 genes in module connectivity. Green Bubble: The genes proven to be involved in low-temperature stress among the top 10 genes in module connectivity. Pink Bubble: Transcription factors among the top 10 genes in module connectivity.)

Supplemental Information 5 Visualization of the interactions of the top 50 genes with the highest connectivity in the turquoise module.

Note: (Blue Bubble: The top 50 genes in module connectivity. Purple Bubble: The top 10 genes in module connectivity. Green Bubble: The genes proven to be involved in low-temperature stress among the top 10 genes in module connectivity. Pink Bubble: Transcription factors among the top 10 genes in module connectivity.)

Supplemental Information 6 Visualization of the interactions of the top 50 genes with the highest connectivity in the m agenta module.

Note: (Blue Bubble: The top 50 genes in module connectivity. Purple Bubble: The top 10 genes in module connectivity. Green Bubble: The genes proven to be involved in low-temperature stress among the top 10 genes in module connectivity. Pink Bubble: Transcription factors among the top 10 genes in module connectivity.)

Supplemental Information 7 Functional annotation of hub genes in the modules associating with low-temperature tolerance.

Supplemental Information 8 Raw RT-qPCR data of Figure 7.

Supplemental Information 9 Gene ontology (GO) enrichment analysis of four modules.

Supplemental Information 10 MIQE checklist.

Supplemental Information 11 Gene pathway (KEGG) enrichment analysis of four modules.

Supplemental Information 12 R code for WGCNA.

Supplemental Information 13 R code for KEGG.

Additional Information and Declarations

Competing Interests

The authors declare that they have no competing interests.

Author Contributions

Chuangye Shi conceived and designed the experiments, performed the experiments, analyzed the data, prepared figures and/or tables, authored or reviewed drafts of the article, and approved the final draft.

Jing Dong conceived and designed the experiments, analyzed the data, prepared figures and/or tables, and approved the final draft.

Chunxiao Zhang conceived and designed the experiments, prepared figures and/or tables, and approved the final draft.

Liquan Sun conceived and designed the experiments, analyzed the data, prepared figures and/or tables, and approved the final draft.

Fengxue Jin performed the experiments, authored or reviewed drafts of the article, and approved the final draft.

Xiaohui Zhou performed the experiments, authored or reviewed drafts of the article, and approved the final draft.

Xueyan Liu performed the experiments, authored or reviewed drafts of the article, and approved the final draft.

Weilin Wu conceived and designed the experiments, authored or reviewed drafts of the article, and approved the final draft.

Xiaohui Li conceived and designed the experiments, authored or reviewed drafts of the article, and approved the final draft.

Data Availability

The following information was supplied regarding data availability:

The original measurement values are available in the Supplemental File.

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
