# Peer review of "Mining of co-expression genes in response to cold stress at maize (Zea mays L.) germination and sprouting stages by weighted gene co-expression networks analysis"

_PeerJ, doi:10.7717/peerj.19124_

## Round 0.1 · original submission · Major Revisions

Include a sentence in the abstract section that relates to the aim of the study.
Include a paragraph in the introduction that describes the genes that play a role in cold tolerance in maize.

It would be a good idea to include the germination percentages you measured in the study in the results.

Reviewer 1 ·

Basic reporting

The article is written clearly in English and can be easily understood. References are sufficient and support the findings. The figures are attractive and clear to reflect the results.

Experimental design

A detailed experimental design is given in the text. The following corrections should be made.

-The budding stage indicates the sprouting stage of germinating seeds. It should be changed to the sprouting stage.
In Line 75, .. at germination stage of two maize inbred lines....
In Line 75, please give the references about whether these maize lines were resistant (R) and susceptible (S) to low temperatures.

Validity of the findings

The findings are valid and useful for researchers.

Reviewer 2 ·

Basic reporting

The written language is understandable but very long sentences are used. It can be written using a more academic language. The number of literature is sufficient and up-to-date literature is used.
No reference to graph 5 in the text. Discussion and conclusion section is sufficient. The image of graph 7 is not clear and the text is too small to read. The references does not comply with the journal's spelling rules.

Experimental design

The research result and experimental design are suitable for the study. The material and method are well explained. Research question well defined, relevant & meaningful.

Validity of the findings

The research results are well explained.

Annotated reviews are not available for download in order to protect the identity of reviewers who chose to remain anonymous.

Reviewer 3 ·

Basic reporting

A recent study has provided groundbreaking insights into the differential gene expressions of cold-sensitive and resistant inbred lines of maize. The research focused on two distinct varieties: "Zhongxi 091/O2," a cold-resistant maize line, and "Chang 7-2," a cold-susceptible maize line. By exposing these varieties to low temperatures, the researchers aimed to identify the genetic factors underlying cold tolerance in maize.

Using transcriptome analysis, the researchers measured the gene expressions in both maize varieties. This approach enabled them to pinpoint specific genes that are differentially expressed in response to cold stress. By comparing the gene expression profiles of the cold-resistant and susceptible lines, the researchers identified potential candidate genes that confer cold tolerance.

However, there are a few concerns that should be addressed before the article could be accepted for publication for example:
Run on sentences e.g Methods part of abstract comprises of two sentences only. Should be split for better readability.
Similarly, Rewrite the second sentence of results section in abstracts as it is exact replica of first sentence except the word budding stage. Similar problem was noted at line 183-191 where the first stage statement sounds exact replica of the second statement where the values are different and stage is different, otherwise it is same or "copy and pasted".
While writing an abstract, it's essential to ensure that this section is clear and concise. However, condensing complex information into a single sentence can lead to run-on sentences that hinder readability.

To improve clarity, it's recommended to rewrite the abstract section again. This allows authors to provide a more detailed description of their methodology, making it easier for readers to understand the research approach.

Experimental design

The Introduction is too short with only two paragraphs, and lacks any survey reports or review articles to highlight the losses incurred due to cold stress in maize. It is suggested that the authors should include the mapping and characterization studies of genes related to cold stress in maize for a meaningful introduction of the problem.
The statement “Maize (Zea mays L.) is one of the most important cereal crops for food, economy, and feed worldwide, accounting for approximately 40% of the total global grain production and is the most widely planted crop.” Over-estimates the situation as the entire maize is not grown in cold areas of the world and the entire crop production is not jeopardized due to cold intolerance.
It also merely reports only two studies in maize for WGCNA analyses? More studies should be reported.

It is unclear why the seeds were dissected at 0 hour interval? As it does not make any sense towards germination?
Same is the case for budding stage as well. What is meant by 0h?

It is unclear how 30 samples were selected for RNA sequencing (line 148) from the selected 200 seeds at germination stage and 100 sprouts from budding stage

Validity of the findings

The authors wrote "Zhongxi 091/O2," a cold-resistant maize line, and "Chang 7-2," a cold-susceptible maize line as varieties in abstract and inbred line in the rest of the article. It is important to clearly describe it either as a variety or an inbred line.

Line 148: A total of 46,987 original reads were obtained from 30 samples.
Line 24: remove full stop after “Methods.” With “:”
Line 31: remove full stop after “Results.” With “:”
Line 44: remove full stop after “crops.”
Line 45: italicize the name “Zea mays”
Line 84: Remove photo from the “0 h light/24 h dark photoperiod” as no light was provided
Line 97: “The total RNA of maize germination stage and germination stage was extracted using” did you mean The total RNA of maize germination stage and budding stage was extracted using the the
Line 102: replace “the end was repaired” with “the ends were repaired”
Line 112: replace “β Calculate” with “the value of coefficient β was estimated by”
Line 138: Super script ® in SYBR ® Green Realtime
Line 180-182: Rewrite as it is very confusing and unclear” During the cold stress treatment of maize at the germination stage, the two materials showed the same trend of differential expression, and the number of germination-resistant (GR) and germination-sensitive materials (GS) increased gradually with prolonged stress time.”
Line 44: remove full stop after “colors.”
Line 296: replace “germination and breeding” with “germination and budding”
Line 304: SA what it stands for? We have to guess if you mean salicylic acid?

Line 325-352: the names of the genes are not italicized???

Line 360: add full stop after process “play in the process These”

---

## Round 0.2 · accepted · Accept

Your manuscript accepted after revisions. Congratulations

Reviewer 1 ·

Basic reporting

It should be published.

Experimental design

Suitable

Validity of the findings

Nice

Additional comments

It should be published.

Reviewer 2 ·

Basic reporting

The introduction section has been expanded to provide a stronger context for the subject and supported by literature. The references section has been updated, increasing the study's compatibility with the literature. The visual elements of the article have also been significantly improved. Tables and graphs have been made clearer and more effective in terms of visualization. These improvements in data presentation have provided clearer interpretation of the results and have allowed the reader to more easily evaluate the findings.

Experimental design

In the materials and methods section, the methods used have been explained in more detail, increasing the reproducibility of the study. Additional analyses and explanations have been added to provide a better understanding of the findings, strengthening the scientific validity of the results.

Validity of the findings

The graphics and tables used were brought into compliance with the relevant standards, and necessary explanations were added to ensure integrity. These arrangements increased the scientific presentation quality of the study and ensured that the article made a stronger academic contribution.
Conclusions are well stated, linked to original research question & limited to supporting results.

Additional comments

The corrections and additional suggestions made during the evaluation process were meticulously addressed by the authors and successfully integrated into the article. The revisions made increased the scientific quality and consistency of the study and made its contribution to the relevant field more apparent. In this context, I think it is appropriate to accept the article.

Reviewer 3 ·

Basic reporting

The English write up improved significantly, and now is quite clear and understandable.

Experimental design

A better explanation of samples and analyses clarified the confusion on the number of samples used for transcriptomic analysis.

Validity of the findings

Looks good